# Design, Manufacturing, and Open-Loop Control of a Soft Pneumatic Arm

Jorge Francisco García-Samartín [ID], Adrián Rieker and Antonio Barrientos *[ID]

Centro de Automática y Robótica (UPM-CSIC), Universidad Politécnica de Madrid-Consejo Superior de Investigaciones Científicas, José Gutiérrez Abascal 2, 28006 Madrid, Spain; jorge.gsamartin@upm.es (J.F.G.-S.); adrian.rieker.gonzalez@alumnos.upm.es (A.R.)
* Correspondence: antonio.barrientos@upm.es

**Abstract:** Soft robots distinguish themselves from traditional robots by embracing flexible kinematics. Because of their recent emergence, there exist numerous uncharted territories, including novel actuators, manufacturing processes, and advanced control methods. This research is centred on the design, fabrication, and control of a pneumatic soft robot. The principal objective is to develop a modular soft robot featuring multiple segments, each one with three degrees of freedom. This yields a tubular structure with five independent degrees of freedom, enabling motion across three spatial dimensions. Physical construction leverages tin-cured silicone and a wax-casting method, refined through an iterative processes. PLA moulds that are 3D-printed and filled with silicone yield the desired model, while bladder-like structures are formed within using solidified paraffin wax-positive moulds. For control, an empirically fine-tuned open-loop system is adopted. This paper culminates in rigorous testing. Finally, the bending ability, weight-carrying capacity, and possible applications are discussed.

**Keywords:** soft actuator; pneumatics; soft robots materials and design; open-loop control; computer vision capture





## 1. Introduction

The rapid growth of soft robotics in the past decade is driven by their versatile applications in various fields, such as assisting surgeons, taking care of elderly people [1,2], aiding rehabilitation [3], and enhancing manipulation capabilities [4]. Soft robots significantly differ from their traditional rigid counterparts, demanding innovative solutions across a wide spectrum, encompassing sensors, actuators, and control techniques.

While conventional actuation methods can be applied in certain scenarios, such as cable robots where external motors tension the cables [5,6], novel actuators are frequently required. This has led to the development of various actuator types, including pneumatic actuators [7], electroactive polymers (EAP) [8], magnetic actuators [9], twisted and coiled actuators (TCA) [10], and shape memory alloys (SMA) [11,12].

Among these options, pneumatic actuators stand out due to their cost-effectiveness, simplified manufacturing, straightforward actuation, and impressive load-bearing capacity [13], especially when no high speeds are required. Certainly, these actuators do not necessitate specific manufacturing conditions beyond the curing of silicone, typically conducted at or near room temperature. Moreover, their operation does not demand high-voltage conditions or the generation of external magnetic fields. Furthermore, they excel in achieving continuous deformations in the robots, surpassing those actuated by wires.

These advantages have led to their use in grippers [14], inspection tasks [15], locomotion robots [16,17], wearable devices for rehabilitation [18], or imitating human body parts [19].

The main contribution of this work has been the design, manufacturing, and open-loop control of a (DOF) soft pneumatic arm with five-degrees of freedom called PAUL

(Pneumatic Articulated Ultrasoft Limb), depicted in Figure 1, capable of carrying light loads without increasing its precision error. In addition to the precision of its control system without and with external payloads, its workspace and its bending capacity are also analysed. It is not common to find a simultaneous analysis of all these factors in the literature.

PAUL is composed of three independent modules or segments, made of silicone. Each segment is fed by three pneumatic tubes whose inflation give him three degrees of freedom: it can bend in two direction, as well as elongate when the three bladders are inflated. To reduce redundancies, it was decided to act only up to two at a time. Combining these components results in a robot with nine inputs and five degrees of freedom in the task space, as no twisting movement is possible in any case. Actuation is achieved by measuring the valve opening duration, while a vision system, based on identifying a trihedron of colours, provides continuous feedback on the position of the end-effector.

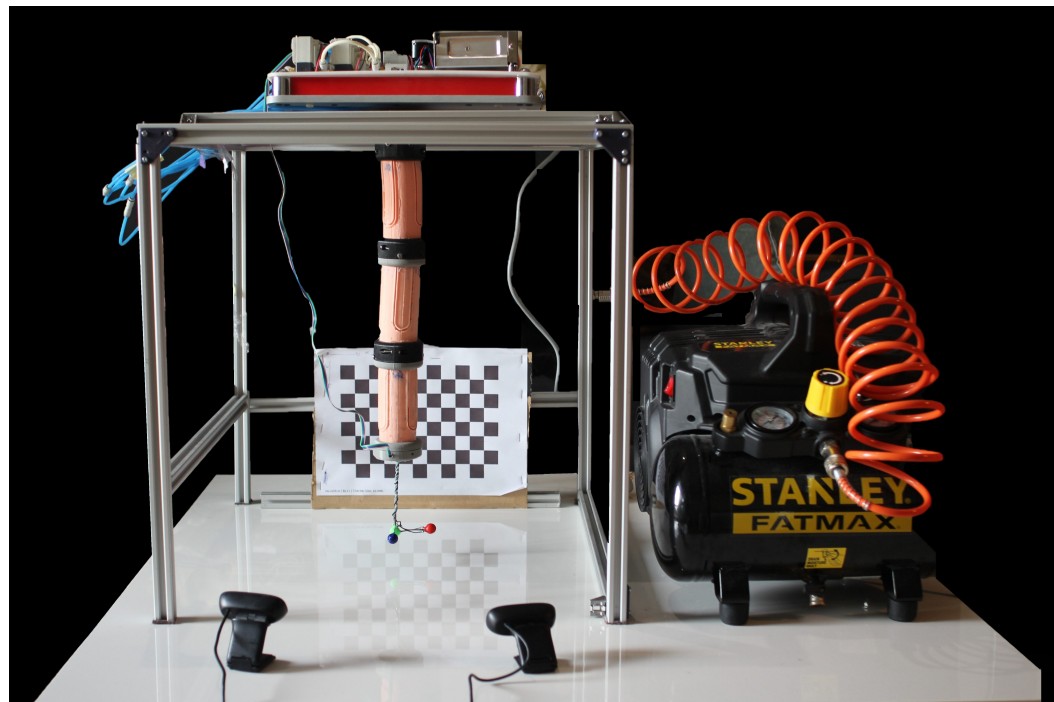

**Figure 1.** PAUL robot. The figure shows the arm, with its three segments and the trihedron that allows its position and orientation to be known, as well as the pneumatic actuation bench (at the top), the compressor (on the right), and the two cameras and the calibration grid, which make up the vision system. Source: authors.

To address the challenge of modelling these robots, especially given the limitations of existing theoretical models—methods like the Piecewise Constant Curvature (PCC), which are adequate for hyper-redundant robots, may lack precision in soft manipulators [20], and parameters for simulations based on the Finite Elements Method (FEM) are difficult to adjust—an open-loop control system has been developed. This system relies on an experimentally constructed table that correlates valve opening times with the corresponding final positions of the manipulator. This empirical approach allows for reasonably accurate modelling of both the direct and inverse kinematics of the robot, even in the absence of precise theoretical models.

This paper is structured as follows: Section 2 presents the main characteristics and working principles of the existing soft pneumatic arms, Section 3 details the PAUL design and manufacturing process, highlighting the various modifications that had to be made iteratively and also addressing the construction of the pneumatic actuation bench. Section 4 describes the open-loop control system and the steps that had to be taken to get there: the configuration of the robot hardware, the implementation of the vision system, and the

generation of the data table. The tests carried out are commented on in Section 5. Section 6 draws the conclusion.

## 2. Related Works

### 2.1. Pneumatic Actuation

There are numerous possibilities for designing and constructing pneumatic actuators. As is common in soft robotics, bio-inspired actuators exist. In [21], an actuator is presented that combines both pneumatics and tendons to mimic and attempt to mimic the behaviour of an octopus arm. Similar to the final operation, although inspired by the human finger, the work of [22] can be considered. This consists of three one-dimensional valves that swell, thus mimicking the movement of the phalanges.

Nevertheless, the most common bio-inspired pneumatic actuators are Pneumatic Artificial Muscles (PAMs). They are based on achieving the extension or contraction of a pneumatic chamber, similar to that of a muscle. The walls of the chamber are usually made of a very thin and flexible membrane, which facilitates large deformations with little air flow introduced [23]. Although some actuators lengthen longitudinally upon inflation [24], the prevalent approach involves utilising McKibben muscles, consisting of a bladder inserted into a braided mesh that constrains the movement of the bladder as it inflates, producing a contraction of the whole [25]. An arm of two PAM segments is designed and modelled in [26].

Another alternative that has gained particular importance in recent years is the Pneunet-type actuators, first introduced in [27]. Manipulators that use this type of action consist of a finned beam-type structure and, in some cases, some material of varying stiffness. It has been widely used for grippers [28,29], soft gloves [30] and for modelling human body parts [31].

Different authors have proposed evolutions to this structure. An optimisation of the geometry is carried out in [32], whereas [33] presents a PneuFlex actuator, which has evolved the Pneunet concept by making the beam have a variable cross-section. On the other hand, the works of [34,35] show how it is possible to design pneumatic segments based on this actuator with bidirectional bending.

Jamming, initially used for soft grippers [36], can also be used to create manipulators in the form of beams [37]. Their function is the reverse of that of Pneunets: in their natural state they are bent and, when negative pressure is applied, they stiffen. In [38], a TPU-printed segment at which pressure or a vacuum can be applied, is introduced.

### 2.2. Pneumatic Arms

In addition to the design of the actuators, pneumatic soft robotics has to face the challenge of their integration for the construction of arms with various degrees of freedom. While some of the previous work, such as [26], does form small arms, several alternatives have been developed for the construction of manipulators.

A first option is the hybrid approaches, in which both rigid and purely soft elements are combined, which makes it possible to obtain a relatively stable mechanism in a simple way. An example of this can be found in [39], in which antagonistically actuated PAM pairs are used to move a rigid beam arm with seven degrees of freedom. A very similar procedure is followed in [40]. To prevent the robot from causing harm to humans, inflatable sleeves are added to the arm.

In [41], a pneumatic segment with rigid bases is developed. This consists of six tubes which, due to their geometry, when inflated in groups of three, allow the assembly to rotate on the axis perpendicular to the bases. Although a whole robotic arm is not developed as such, but only a segment of one degree of freedom, it is integrated into an entirely soft robotic arm. The same authors had previously developed a six-degree-of-freedom robotic arm from 85 cm-long segments, capable of lifting loads of up to 3 kg [42].

The work of [43], on the other hand, presents an origami robotic arm made of TPU. This is inflated and deflated by pneumatic actuation, but its position is controlled by tendons

which, while slowing down the system, make it much more precise. Among 3D-printed robots, the authors of [44] present a three-degree-of-freedom segment, whose movement is achieved by inflating or deflating one or more of its three pneumatic tubes. It also has cables that play the role of antagonist and stiffen the movements of the manipulator. Its working principle is very similar to that of the work in [1].

Also based on the philosophy of printable and deployable robots, the authors of [45,46] present a Honeycomb Pneumatic Network (HPN) arm. It has been constructed by concatenating TPU honeycomb structures, each with an airbag inside. Several prototypes are presented in the paper, one of which can reach a length of 600 mm after joining four segments together. While several of its advantages are discussed, it presents the problem that its weight, without being exaggerated, is high: the arm, considering all the tubes, weighs 4.4 kg.

The robot in [47] consists of two segments, each with three pneumatic tubes. Although this theoretically gives it six degrees of freedom, the controllers are only responsible for the upper module, which does not allow all positions and orientations to be freely fixed.

The STIFF-FLOP manipulator was introduced in [48]. This consists of an elastomeric cylinder with a series of pneumatic chambers inside, the inflation and deflation of which causes deformation of the cylinder and, therefore, movement of the robot. Various iterations of this design exist, such as the STIFF-FLOP segment with stiffening tendons demonstrated in [31].

In this line, SoPrA, presented in [49], is made combining three fibre-reinforced silicone segments, each shaped like a conical trunk, so that the end of the robot is much narrower than the base. Although the truncated cone shape is advantageous because, as the authors point out, the upper segments require more torque and hold more tubes inside them, the manufacturing process used to achieve the taper prevents new segments from being easily added to the robot.

In [50], a three-segment arm is built using silicone rubber. Each segment is 110 mm long, has a diameter of 45 mm and, contrary to traditional STIFF-FLOP structure, is equipped with 4 inflatable cavities. This implies an increase of weight and difficulty of control, as redundancy is increased compared to having only three degrees of freedom per segment.

### 2.3. Control of Soft Robots

If in the design of pneumatic soft robots there is a wide variety of possibilities due to the still immature state of these robots, in control the possibilities increase even more. Indeed, controlling soft robots is still an open challenge [51] and there are many possible solutions, even if they do not yet achieve similar precision to those existing in rigid robotics.

Both model-based and data-driven control methods have been tested. Although some authors suggest that, unlike what has happened in other areas, where it started with model-based controllers and has evolved to the use of Machine Learning techniques, here the opposite is happening [52]. For the moment, both techniques coexist and it does not seem that, in the short term, either philosophy is going to impose itself.

Model-based techniques include the use of PCC on the one hand and FEM on the other. The former has been used successfully in pneumatic soft robots [50,53]. Its main drawback, however, is that it is based on very strict premises—deformation with constant curvature and absence of gravity—which often make its application unfeasible.

The use of FEM, on the other hand, can achieve very good results. As pointed out by [52], although numerical methods are theoretically less accurate than analytical methods, when modelling soft structures such as these, analytical methods are not capable of working with such complex—or sometimes simply unknown—shapes, constitutive laws, and boundary conditions.

Its use has been widely implemented in soft robots [12,54]. In the case of pneumatic soft robots, the work of Ding [55]—which is, however, much smaller in size than PAUL—or Cangan [56], which uses previously presented SoPrA arm, stands out. The prob-

lem they present, however, is the high computational cost required, which often makes their closed chain control unfeasible, unless reduced-order models are used [57]. Furthermore, setting the different elastic and geometric parameters of the materials is not an automatic task due to how difficult it can be to characterise them.

Thus, different Machine Learning (ML) techniques are usually used to solve the modelling and control problem. Many techniques have been used, from Feedforward Neural Networks (FFNN) [45] to more complex network architectures [25,58] in addition to the use of Reinforcement Learning [59,60]. A wide variety of input data can be used to generate a fairly accurate model: data-driven models have been tested to perform well with input from the real robot [61], data from a FEM model [12], a combination of both [62], and visual data [63].

No relationship has been observed between the complexity of the ML technique used and the results obtained. On the contrary, the philosophy in these cases is to always use the simplest tool possible—which usually also requires a smaller amount of data—that achieves the expected results. Although it is not common, controls based on polynomial adjustments [5] or minimisation of cost functions [64] have even been achieved. This is the philosophy that has been followed with PAUL. The objective, furthermore, in this first stage, is not to achieve very accurate control—the implementation of sensors in it is expected as future work—but to have a model that allows carrying out initial movement tests.

## 3. PAUL: Design and Manufacturing

### 3.1. Robot Design

Before starting the design and subsequent manufacture, some dimensional and geometrical specifications imposed by the functionality sought have been set. These requirements can be summarised in the following points:

- The resulting robot must consist of three independently actuated segments, each with three degrees of freedom.
- The actuation of the segments that make up the robot must be pneumatic.
- The segments must be made of flexible silicone.
- These segments must allow easy assembly and disassembly as well as a modular design.
- The pneumatic tubes must be completely embedded in the body of the robot to avoid breakage and to allow more complex movements.

It was decided that each segment should be cylindrical, 100 mm high and 45 mm in diameter, with three pneumatic bladders evenly distributed along the cylinder. In order to fix the height of the cylinder, other existing robots in the literature were used as references [12,50] in order to achieve comparable working spaces. The diameter, on the other hand, was to be as small as possible—in order to make the robot as light as possible—but at the same time to allow all the tubes to pass through the inside of the segment. In addition, it had to have a sufficient wall thickness between the bladder and the outside to prevent ruptures during inflation and deflation. The final size was decided after several design iterations and three manufacturing tests.

Bladders, which are depicted in Figure 2, have a Pneunet structure. As was the case when designing the rest of the segment, some CAD iterations were carried out before a final geometry was fixed. In particular, we chose to use as rounded edges as possible to reduce the stresses and, therefore, the probability of punctures, during the numerous inflations and deflations. In addition, different values for the number of protrusions in each bladder were considered, finally using nine, as it was seen that a higher number made the deformation of the segment more curved when it was inflated, which can facilitate its modelling using PCC.

As there are three bladders per segment, each segment should have three degrees of freedom; however, it was resolved to inflate only one or two simultaneously to reduce the redundancy of the system, making control easier.

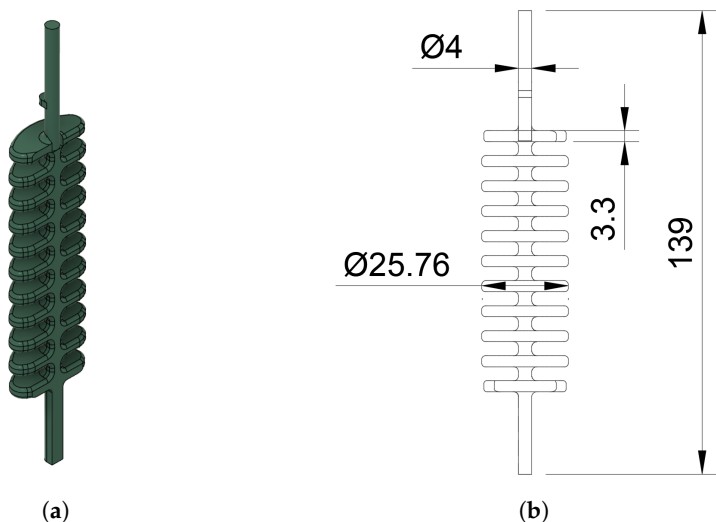

(**a**)  (**b**)

**Figure 2.** Bladder cores. (**a**) CAD Model. (**b**) Dimensioned drawing. All dimensions are in mm. Source: authors.

Final design of PAUL's segments is shown in Figure 3. Each segment has three U-shaped grooves on its surface, designed for the subsequent insertion of elastomeric sensors to measure the deformation experienced by the robot when it inflates and thus predict the position of the robot and/or the times that each bladder has been actuated.

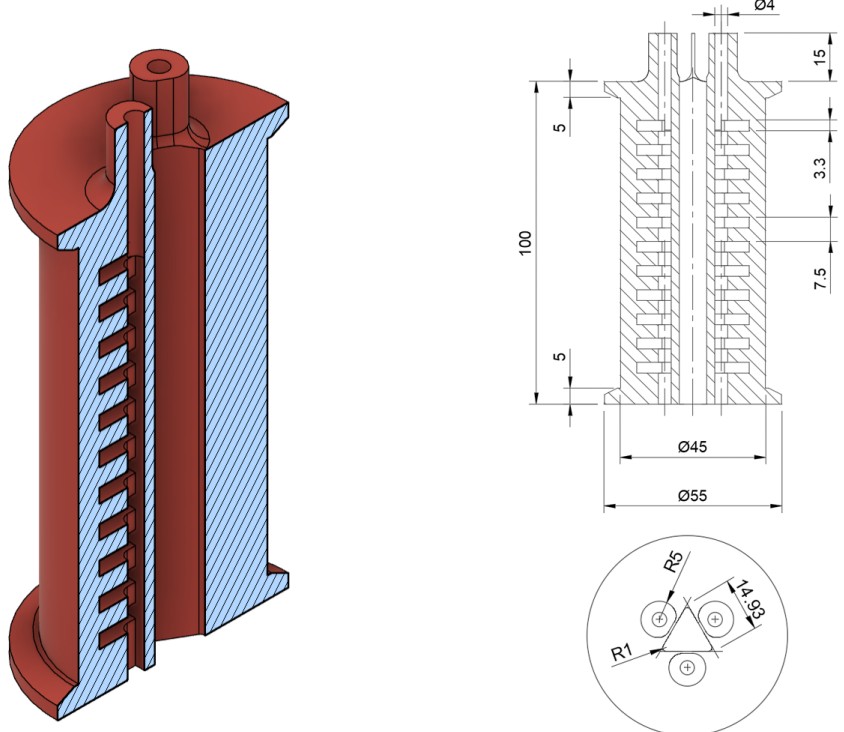

**Figure 3.** Final design of PAUL's segments. All dimensions are in mm. Source: authors.

It was decided that the connection between segments would be made using 3D-printed devices, as the ones presented in Figure 4, each 200 mm high. While PLA is not a completely soft material, some soft robots incorporate parts with a degree of stiffness within their overall soft structure [50,55,65]. This choice enhances PAUL's modularity as the connections can be easily established. It is enough to attach a new segment and insert the tubes—and the cables, when sensors will be implemented—through the central hole of the previous ones, eliminating the need for adhesives or sealants.

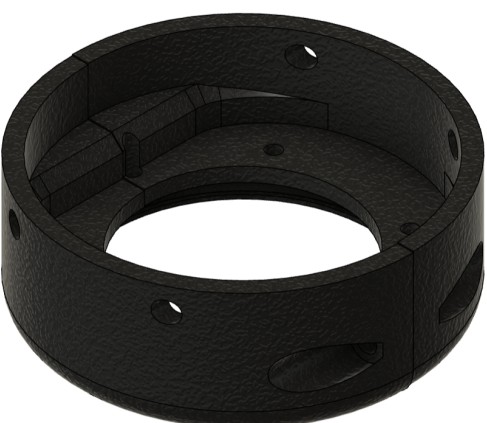

**Figure 4.** 3D-printed connection between segments. Source: authors.

In order to achieve optimum adhesion of the segments to the printed parts, a larger diameter section has been incorporated at the end, increasing in consequence the adhesion surface. Likewise, in order to achieve a better seal, the pneumatic tubes are not inserted directly on the bladders, but on some previous projections on which plastic flanges will be fitted to reinforce this union and prevent accidental leaks.

*3.2. Material Selection*

Due to the pneumatic operation of the robot, it was essential to find a material that combined flexibility to deform with air and the ability to maintain a reliable seal to prevent air leaks. Therefore, silicone was chosen for the segments, and 3D-printed connectors were employed to link them, as explained in the preceding section.

The silicone segments were fabricated through a moulding process, with the moulds being 3D printed on an Artillery Genius Pro printer. The printer was configured using the parameters detailed in Table 1. The same table provides the manufacturing parameters for the connectors.

**Table 1.** 3D printer parameters.

| Parameter | Mould | Connectors |
|---|---|---|
| Layer Height | 0.1 mm | 0.2 mm |
| Infill | 5% | 14% |
| Number of Perimeters | 2 | 3 |
| Extrusion Temperature | 195 °C | 195 °C |
| Bed Temperature | 200 °C | 200 °C |

Three different silicones were tested throughout the manufacturing process: PlatSil FS10, EasyPlat 0030, and TinSil 8015, whose characteristics can be compared in Table 2. The first silicone, PlatSil FS10, was rejected as its low curing time results in the appearance of bubbles in the final segment. These acted as stress concentrators, causing damage and air leakages after a few working cycles. The second silicone, EasyPlat 0030, was finally discarded as its low hardness required very thick-walled segments if leakage was to be avoided, which inevitably entailed a high weight.

TinSil 8015, a tin-cured silicone, was selected as it produced segments with an optimal balance between durability and weight. However, it has two drawbacks. First, it is highly toxic, which necessitates the use of special protective measures when working with it, and makes it impossible for the segments to cure in a human traffic area. On the other hand, it shows contractions during the curing of 1% due to the production of alcohols, necessitating a complete filling of the module to counteract this effect.

**Table 2.** Parameters of the different tested silicones.

| Property | PlatSil FS10 | EasyPlat 0030 | TinSil 8015 |
|---|---|---|---|
| Type | Platinum | Platinum | Til |
| Shore Hardness | A13 | 00-30 | A15 |
| Curing Time | 12 min | 4 h | 24 h |
| Viscosity | 4.2 Pa s | 3 Pa s | 12 Pa s |
| Density | 1.08 kg/m$^3$ | 1.04 kg/m$^3$ | 1.1 kg/m$^3$ |

### 3.3. Manufacturing

The first step in the manufacturing process is to obtain the wax cores which, when inserted into the mould, are used to create the holes for what, in the finished segment, will be the bladders. These are made by pouring paraffin wax into previously made female moulds (Figure 5a).

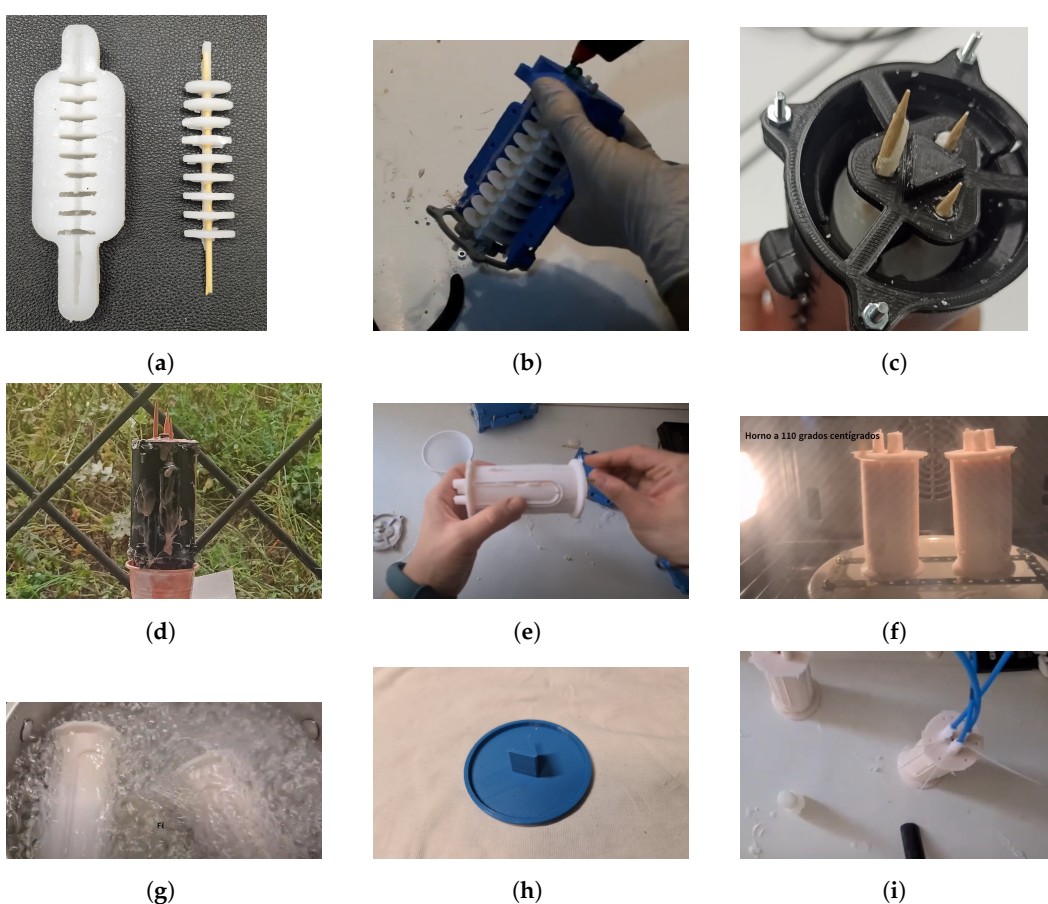

**Figure 5.** Complete PAUL manufacturing process. (**a**) Bladder manufacturing. (**b**) Mould assembly. (**c**) Mould assembled with the three bladders in place, ready for pouring the silicone. (**d**) Curing of the silicone. (**e**) Removal of excess parts. (**f**) Melting of the wax in the oven. (**g**) Bathing of the segment in boiling water. (**h**) Sealing the bottom of the mould. (**i**) Placement of the tubes. Source: authors.

After half an hour, the wax has solidified and the cores can be removed and inserted into the mould (Figure 5b). The mould consists of four 3D-printed parts (two sides, a bottom cap, and a top grip on which the cores rest), which are screwed together and then sealed with a hot silicone bead to prevent leakage during subsequent curing (Figure 5c).

The silicone can then be poured into the mould, which must be filled to the top to counteract the aforementioned shrinkage. In particular, TinSil8015 requires a mass ratio of 10:1 liquid to catalyst. For the dimensions of the segment, about 175 g of total mixture is required.

The curing process lasts 24 h at ambient temperature (Figure 5d), after which it can be removed from the mould. It may be necessary to use a scalpel to remove the silicone burrs (Figure 5e).

Once the segment has been built, the cores that have been used to create the bladders are removed. While the wood can be removed by pulling, it is necessary to apply heat to the segment to remove the wax. Thus, it is first placed in an oven at 110 °C (Figure 5f) and then immersed in a boiling water bath for 15 min, which ensures the elimination of the residual traces of wax (Figure 5g).

Since the males are through, it is required to close the lower part of the segment. To complete this, a layer of silicone is poured onto the plate of Figure 5h, glued onto the segment and left to cure for 24 h. Finally, the pneumatic tubes are joined to the segment, adhering them with cyanoacrylate and strengthening the tightness with the usage of plastic flanges (Figure 5i).

The final result, a functional segment is depicted in Figure 6. Experimentally, it is found that its weight is 161 g and that, as designed, it has a height of 100 mm and an external diameter of 45 mm.

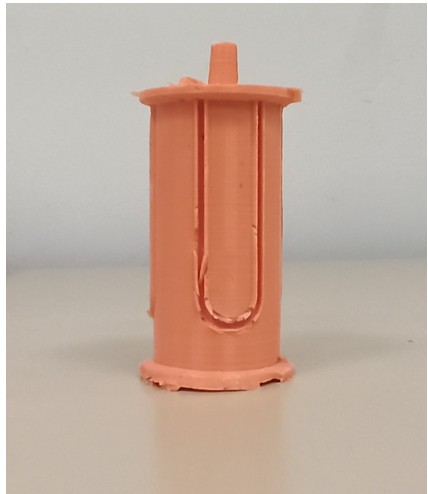

**Figure 6.** Final segment. Source: authors.

*3.4. Actuation Bank*

Within the robot, the function of the pneumatic bench is to control the flow of compressed air from the compressor according to the control signals. Specifically, the PAUL bench consists of 6 pairs of 2/2 valves (SMC VDW20BZ1D model) and 3/2 valves (SMC Y100 model) placed in series, which will therefore allow up to 12 degrees of freedom. Both are shown in Figure 7. The physical characteristics of the 2/2 valves limited the total pressure of the assembly to 4 bar, but to reduce the risk of segment leakage, it was reduced with a flow regulator to 2 bar. Figure 8 presents a schematic of the pneumatic circuit.

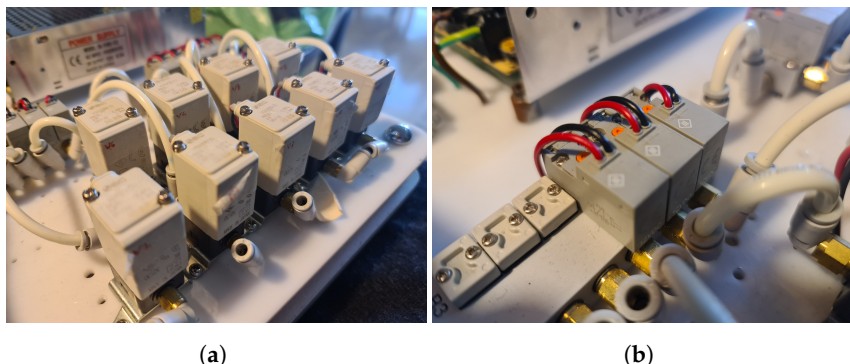

(**a**)                                    (**b**)

**Figure 7.** Valves of PAUL actuation bench (**a**) 2/2 valves. (**b**) 3/2 valves. Source: authors.

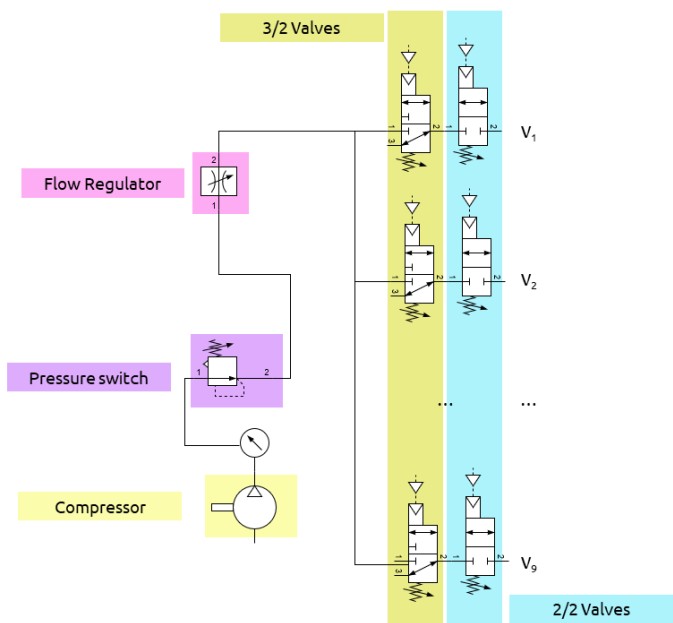

**Figure 8.** Schematic of the pneumatic circuit. Source: authors.

The valves are operated via 24 V voltage signals. A MOSFET (model IRF540) is the switch in charge of managing them. Initially, the use of relays was considered, but the high current they would consume made their use unfeasible. An Arduino Due was chosen as the bench controller. A PC power supply, capable of supplying up to 8.5 A, is responsible for powering the unit, whose final layout is illustrated in Figure 9.

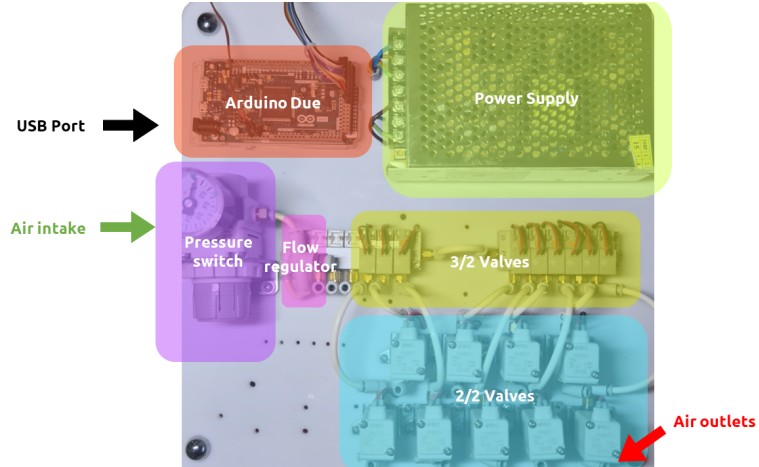

**Figure 9.** Final layout of the pneumatic bench. Source: authors.

## 4. Data Acquisition and Open-Loop Control

### 4.1. Hardware Setup

In order to provide the manipulator with a solid and stable fastening system, which would also allow reliable and predictable data capture of the positions and orientations of its end, the metal structure shown in Figure 10 was built. It is a cube made of steel profiles with methacrylate sheets on the walls. The pneumatic bench, the power supply and the microcontroller were placed on top of the structure.

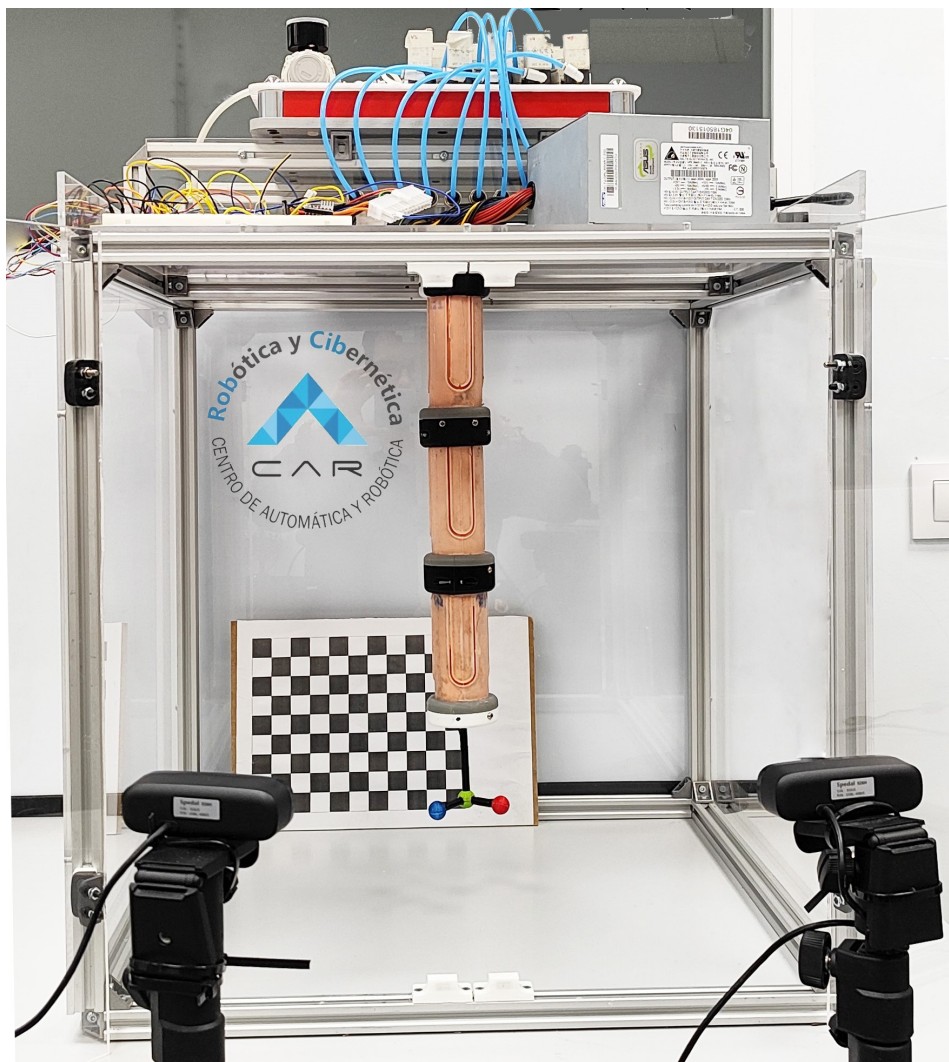

**Figure 10.** PAUL and its working environment. The manipulator can be seen, with three segments inside the box that contains it, on top of which the pneumatic bench and the power source have been placed. At the end of PAUL is placed the trihedron that allows the positions to be captured with the cameras seen in the foreground. Source: authors.

The aim of the data acquisition system is to be able to measure, whenever required, the position and orientation of the end of the robot in order to be able to relate it to the inflation times of each bladder and thus be able to create an open-loop model of PAUL. For this purpose, three elements are available: the cameras, the calibration grid and the trihedron.

Two Spedal AF926H USB cameras with 1920 × 1080 px, 80° field of view and a frequency of 60 fps are used to capture the images. These have been placed on two tripods external to the robot's structure. They are calibrated with a checkerboard of 11 × 8 squares of 20 mm each, which can be seen in Figure 11a.

The vision beacon, on the other hand, has the task of being recognised in space to determine the position and orientation of the mobile system with respect to the fixed system. The trihedron, displayed in Figure 11b, consists of three spheres, manufactured by 3D printing in PLA, inside which three LED diodes have been embedded. Thanks to these, it is possible to vary the luminosity of the spheres by means of software, keeping the system functioning correctly when the workplace or the environmental or lighting conditions vary.

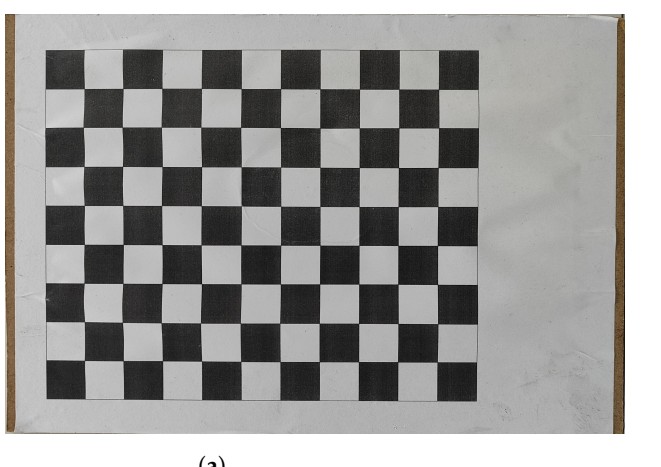
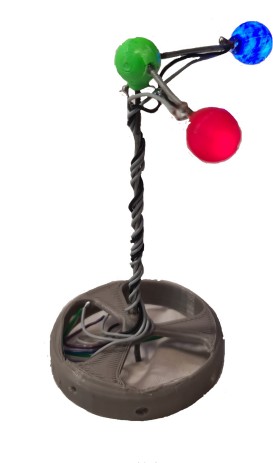

(**a**)       (**b**)

**Figure 11.** Elements of the vision system. (**a**) Calibration grid. (**b**) Beacon to allow the capture of position and orientation. Source: authors.

The existence of the central rod, which moves the luminous spheres away from the base of the robot end, makes possible the spheres to be visible to the cameras in all the poses that the robot can adopt. If the spheres were otherwise directly attached to the end of the robot, there would be numerous poses in which it would not be possible to determine the position, as the spheres would be hidden by the robot itself.

*4.2. Vision Capture System*

MATLAB R2022b version and the Computer Vision and Image Processing toolboxes are used to manage the entire vision system at the software level. The function of the vision system is to obtain the position and orientation of the beacon in the coordinates of the reference system of the real world ($S_w$) from the LEDs positions detected by each of the cameras in their own two-dimensional reference system ($S_c^1$ and $S_c^2$) [66]. The three reference systems are displayed in Figure 12.

In a first step, each camera detects, on its side, the positions of the green, red, and blue spheres, denoted as $(u_g^i, v_g^i)$, $(u_r^i, v_r^i)$, respectively, and $(u_b^i, v_b^i)$, where $i$ indicates the camera in use. To achieve this, a filter is firstly applied to the image, in order to remove the pixels below a specified colour threshold. Despite the seemingly counter-intuitive choice, as the beacon is a highly luminous object that stands out visually from its surroundings, the HSV color space yields better results at this stage compared to RGB.

Subsequently, the filtered pixels are grouped into Maximally Stable External Regions (MSER), which are elliptic areas with similar colour intensities. If multiple MSER regions are detected, the one with lower eccentricity is considered as corresponding to the beacon sphere.

Relationship between detected position, in pixels, and real-world position of the sphere of colour $k$, denoted by $r_k = (x_k, y_k, z_k)$, is obtained, for camera $i$, using Equation (1):

$$Z_c^i \begin{bmatrix} u_k^i \\ v_k^i \\ 1 \\ 1 \end{bmatrix} = M^i H^i \begin{bmatrix} x_k \\ y_k \\ z_k \\ 1 \end{bmatrix} \tag{1}$$

where:

- $Z_c^i = \begin{bmatrix} z_c^i & z_c^i & z_c^i & 0 \end{bmatrix}$ corresponds to the depth of the camera (the distance from camera to the detected sphere), which is unknown.

- $M^i = \left[\hat{M}^i_{3\times3} \middle| \mathbf{0}_{3\times1}\right] = \begin{bmatrix} f_x & 0 & c_x & 0 \\ 0 & f_y & c_y & 0 \\ 0 & 0 & 1 & 0 \end{bmatrix}$ denotes the camera intrinsic parameters

  matrix, on which the focal lengths ($f_x$ and $f_y$) and offsets ($c_x$ and $c_y$) are reflected. These parameters are specific to each camera. In order to obtain them, it is necessary to perform some kind of intrinsic calibration the first time the camera is used. In this case, we have used the default calibration in MATLAB, which consists of taking several pictures of the chessboard in Figure 11a at different angles and then calculating the distortion in each one of them. Latest null column has been inserted in the matrix to fit with the dimensions of $H^i$.

- $H^i = \begin{bmatrix} R^i_{3\times3} & t^i_{3\times1} \\ \mathbf{0}_{1\times3} & 1_{1\times1} \end{bmatrix}$ contains the rotation matrix ($R^i_{3\times3}$) and the translation vector

  ($t^i_{3\times1}$) from the real-world system to the camera system. As this matrix depends both on the position of the coordinate origin and on the position of the camera, which can easily be moved due to accidental slippage, it is necessary to recalibrate it at the beginning of each working session. For this purpose, the extrinsic calibration protocol, also available by default in MATLAB, is used, and the grid in Figure 11a, whose lower-left corner is taken as the origin of the real-world reference system. From the dimensions of the squares, the translation and rotation with respect to their reference frame can be estimated. Subsequently, in the first measurement, the green sphere is taken as the origin of the real-world reference frame.

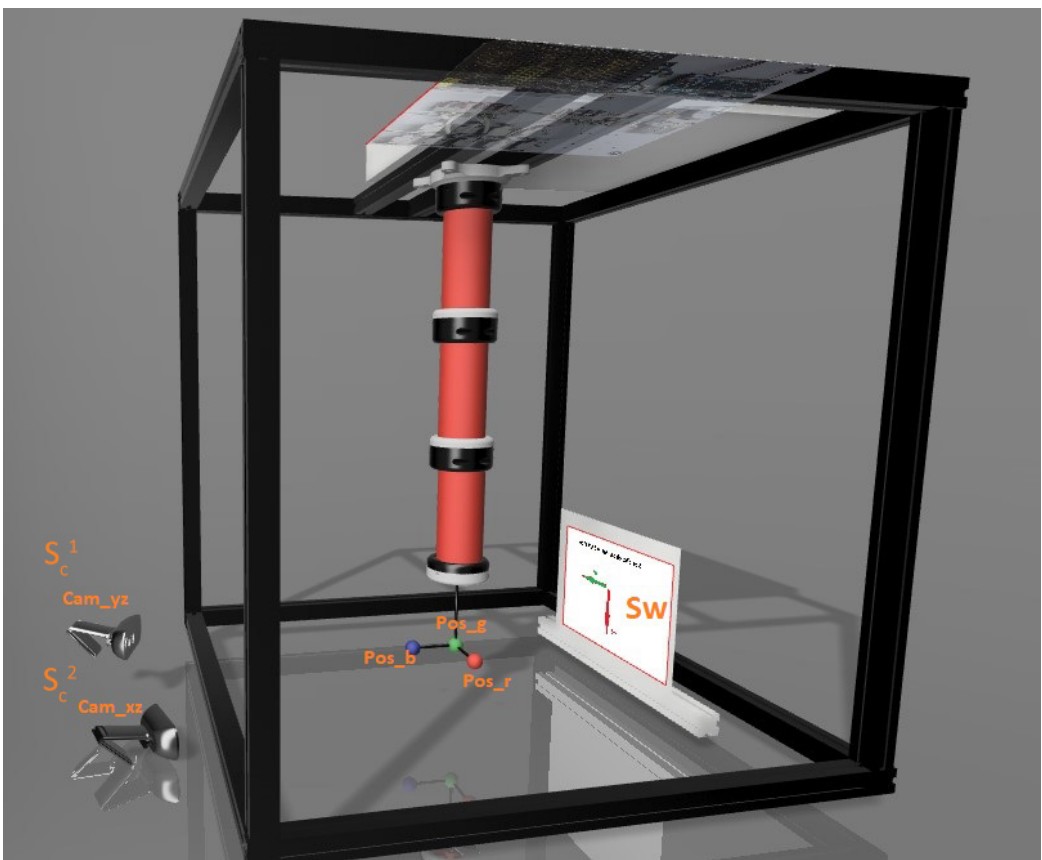

**Figure 12.** PAUL CAD model showing the reference systems of the cameras ($S_c^1$ and $S_c^2$), the real world ($S_w$), and the position of the three LEDs used to determine the position and orientation of the end of the manipulator ($Pos_r$, $Pos_g$ and $Pos_b$). Source: authors.

Because coordinates of the real world are independent of the camera, if Equation (1) is applied for both cameras and $r_k$ vector cleared in the two equations, it can be said that

$$A \begin{bmatrix} z_c^1 \\ z_c^2 \end{bmatrix} = B \tag{2}$$

where

$$A = \left( \hat{M}_{3\times 3}^1 R_{3\times 3}^1 \right)^T \begin{bmatrix} u_1 \\ v_1 \\ 1 \end{bmatrix} - \left( \hat{M}_{3\times 3}^2 R_{3\times 3}^2 \right)^T \begin{bmatrix} u_2 \\ v_2 \\ 1 \end{bmatrix} \tag{3}$$

$$B = \left( \hat{M}_{3\times 3}^1 R_{3\times 3}^1 \right)^T \hat{M}_{3\times 3}^1 t_{3\times 1}^1 - \left( \hat{M}_{3\times 3}^2 R_{3\times 3}^2 \right)^T \hat{M}_{3\times 3}^2 t_{3\times 1}^2 \tag{4}$$

System of Equation (2) can be solved using the Least Squares Method:

$$\begin{bmatrix} z_c^1 \\ z_c^2 \end{bmatrix} = (A^T A)^{-1} A^T B \tag{5}$$

By entering the value of $z_c$ in Equation (1) for either of the two chambers, the real world coordinates of the identified sphere can then be obtained.

Once the coordinates of the three spheres ($r_g$, $r_r$ and $r_b$) have been obtained, the position of the end of the robot is available, but not its orientation. This can, however, be easily obtained. It is sufficient, first of all, to normalise the vectors

$$p = \frac{r_r - r_g}{\|r_r - r_g\|} \tag{6}$$

$$q = \frac{r_b - r_g}{\|r_b - r_g\|} \tag{7}$$

and then use the Rodrigues' rotation formula to obtain it, respect to the real-world base in the form of a rotation matrix:

$$R_w = I + \Omega + \frac{1 - \langle p, q \rangle}{\|\omega\|^2} \Omega^2 \tag{8}$$

where

$$\omega = p \times q \tag{9}$$

$$\Omega = \begin{bmatrix} 0 & -\omega_3 & \omega_2 \\ \omega_3 & 0 & -\omega_1 \\ -\omega_2 & \omega_1 & 0 \end{bmatrix} \tag{10}$$

and $I$ denotes the identity matrix of size 3.

### 4.3. Dataset Generation: Table-Based Models

Due to the complexity of the robot, model-based methodologies, such as PCC or the ones based on Cosserat Rod Theory were discarded. Although the usage of FEM is an avenue that will not be closed in future work, the large number of parameters to be set experimentally for each segment (Young's modulus, moment of inertia...), given that the manufacturing process is so variable meant that, in this first phase, we opted to use some type of PAUL modelling based on data collection.

The output of the system is taken as the position and orientation reached by the final end—thus, at this stage, all the positions of the intermediate segments are ignored—and as input, the inflation times of each of the bladders. As there were not enough pressure sensors available at the time of the construction of the robot, it was decided to take inflation time as an input variable. As the working pressure is limited by the pressure limiting valve

and the flow rate into each bladder can be assumed to be constant, the time is equivalent to the volume of air introduced into each cavity.

All the control options considered have in common the need for a large amount of empirical data, which leads to the need to develop an experimental design to systematise the collection of this data. Since the capture of this information is performed in different phases and the datasets have to represent the behaviour of the robot in an objective way, the re-applicability of the experiment takes on special importance.

The data stored in the datasets was the position of the robot tip and the set of inflation times that achieve this configuration. The aforementioned limitation that only two of the three bladders in the segment are inflated reduces redundancies. As previously stated, more than two segments lead to redundancies, which implies that the inverse kinematic model of the robot can have multiple solutions.

The data collection process involves several sequential steps. Initially, a set number of samples is determined. For each sample, MATLAB commands dispatch a random combination of nine inflation times, corresponding to each valve of PAUL, to the actuation bench. Times are generated below a maximum time limit $T_{max}$, and ensuring that only two cavities per segment are inflated. Following this, the robot's bladders are inflated based on the sent times. Subsequently, the vision system's two cameras capture images to determine the position and orientation of the robot's end. This entire procedure is repeated for the specified number of iterations, and upon completion, the collected data are stored in the dataset.

The information on swelling times is stored as a percentage, with a value of 0 corresponding to a zero swelling of that segment and 100 corresponding to $T_{max}$, the swelling for the maximum number of milliseconds defined for this data collection session. This value $T_{max}$ is stored, together with the values, in the dataset, in order to be able to compare different datasets. The reason for this coding comes from the lack of information, a priori, on what is the maximum pressure that a PAUL bladder admits. Although it is true that it was experimentally determined that inflation times of more than 1500 ms in a row led to punctures, the application of lower times during a repeated number of cycles also generated leaks. On this basis, it was decided never to inflate any valve, either in one or several steps, more than 1000 ms.

Along with the inflation times of each bladder, the position and orientation reached by the end tip is stored, based on the camera readings. In particular, the position of the green marker and the orientation of the trihedron are stored. The latter is expressed in Euler angles, as it is a much efficient form of storage than a rotation matrix. In addition, the dataset also contains metadata from the collection process that are believed to influence the results, such as the pneumatic line pressure or the ambient temperature.

Some aspects in the pneumatic system merit attention. Initially, bladder inflation and deflation are not symmetrical processes. Geometric constraints in the pneumatic components result in a lower deflation rate compared to inflation. Consequently, when the PAUL receives a deflation time, it multiplies it by an empirically derived factor, approximately 1.45 for a 1.2 bar working pressure. This multiplier compensates for the discrepancy between inflation and deflation times of a singular group of bladders, ensuring that the deflation time aligns with the time required to reach the same inflation point.

Similarly, although it is physically possible to inflate several valves at the same time, it has been shown that this parallel flow distribution means that the effective fillings of each valve are not the same as if they were inflated individually. To prevent this phenomenon, it was decided to inflate each bladder individually both during the data acquisition process and utterly, when PAUL was asked to reach certain positions.

Finally, there are hysteresis phenomena in the silicone that cause the position reached by inflating for a time $t$ to be different from the position reached by inflating first for a time $t_1$ and then for a time $t_2 = t - t_1$. The strategy employed to tackle this problem was to capture the dataset bringing PAUL back to its zero position between each sample. Nevertheless, when controlling the robot in open-loop this is not possible, or, at least,

not desirable, as one may wish to follow trajectories or travel through a sequence of points. Therefore, transitioning from position $x1$ to $x2$ requires an additional factor of 1.2, also derived experimentally, to account for hysteresis effects.

### 4.4. Open-Loop Control

Once the dataset is generated, it can be used to model the behaviour of PAUL for open-loop control. It is foreseen, as a future line, to train a neural network for the direct kinematics and another one for the inverse kinematics. However, given the large amount of data that may be required (in [62], 24,389 samples are used for a three-segment robot like this one), a table look-up method has been used for this work.

The method for direct kinematics—which allows obtaining the position and orientation of the final end of the robot from the inflation times of the nine bladders—consists of searching, in the generated dataset in the previous step, the three inflation time values located at a shorter distance from the inflation time given as a reference. Obviously, if the set of inflation times sought were in the table, the value associated with these times would be returned as a result of the direct kinematic model. Otherwise, the average of the position and orientation values associated with the three closest inflation times, weighted by the distance (Euclidean norm) existing between each of them and the values of reference inflation times, is returned as the position and orientation value of the robot.

Mathematically, given an input $t \in \mathbb{R}^9$, with $t_1$, $t_2$ and $t_3$ being the three closest inflation times collected in the dataset and $x_1$, $x_2$, and $x_3 \in \mathbb{R}^6$, their respective stored positions during the data collection process, firstly, the weighs that will be used for the weighting are calculated:

$$\alpha_1 = \frac{1}{\|t - t_1\|} \tag{11}$$

$$\alpha_2 = \frac{1}{\|t - t_2\|} \tag{12}$$

$$\alpha_3 = \frac{1}{\|t - t_3\|} \tag{13}$$

with them, it is possible to calculate the position returned by the direct kinematic model using the following expression:

$$x = \frac{\alpha_1 x_1 + \alpha_2 x_2 + \alpha_3 x_3}{\alpha_1 + \alpha_2 + \alpha_3} \tag{14}$$

For the inverse kinematic model, the position values closest to $x$ are searched in the table and the weighted average of their associated inflation times is returned. In this way, the weight $\hat{\alpha}_i$ values are calculated with the following expression:

$$\hat{\alpha}_i = \frac{1}{\|x - x_i\|} \qquad i = 1, 2, 3 \tag{15}$$

and, finally, the necessary inflation times using

$$t = \frac{\hat{\alpha}_1 t_1 + \hat{\alpha}_2 t_2 + \hat{\alpha}_3 t_3}{\hat{\alpha}_1 + \hat{\alpha}_2 + \hat{\alpha}_3} \tag{16}$$

## 5. Results

### 5.1. Final PAUL Version

The version of PAUL we decided to work with for the experiments has three segments. As each segment has 2 degrees of freedom, the robot as a whole has $2^3 = 8$ degrees of freedom in a 6-dimensional space, which implies a degree of redundancy of 2.

Although the layout of the pneumatic bench allows working with up to four segments, it was thought that using three would allow the different problems linked to

redundancy to be tackled without increasing the weight of the robot too much or requiring the tubes—which pass through the interior of the segments—to have an excessive amount of space.

It is true that the tubes of the other three could pass through the first module, nevertheless, it was thought that the stiffness they would introduce by being so compressed could make it difficult to bend the initial segment. Since it is also the segment that has to exert the most force, as it is the one that supports the weight of the other segments, the risk of punctures could be increased.

Therefore, a robot consisting of three identical modules was assembled, standing at a total height of 390 mm (with each segment measuring 100 mm, intersegment connections 20 mm each, and the vision trihedron rod 30 mm). Under these configurations, the estimated weight of PAUL's arm is around 600 g. The structure protecting the manipulator is a cube with a side of 500 mm. Pressure of the pneumatic line was established in 1.2 bar.

Examples of PAUL reaching different positions are depicted in Figure 13.

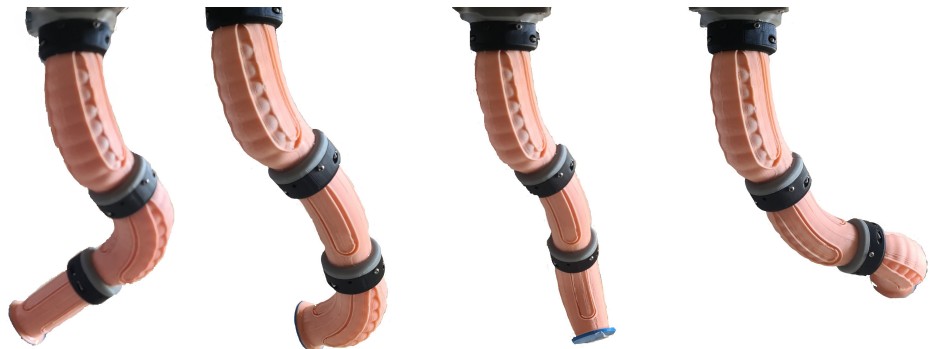

**Figure 13.** PAUL presented in various poses. Source: authors.

*5.2. Workspace Analysis*

The analysis of the workspace has been carried out experimentally, based on the data taken to generate the dataset. Figure 14 shows the workspace of a segment.

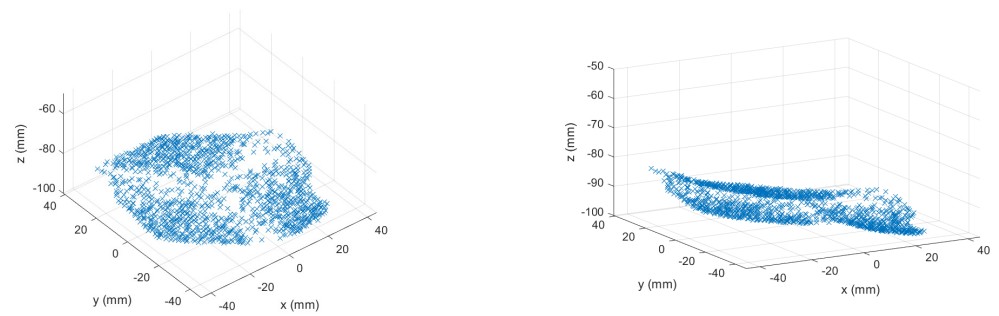

**Figure 14.** Workspace of a single segment, viewed from two different angles. It consists on three spherical surfaces intersecting in the centre, which corresponds to the initial point of the segment, when no air has been sent to the bladders. Source: authors.

As can be seen, this is a surface, as the segment has two degrees of freedom if the condition that at least one valve should remain deflated is imposed. The surface can be considered as the union of three surfaces intersecting at the central point, which corresponds to the configuration of all deflated bladders. The three surfaces are roughly spherical in shape. If the PCC model were completely valid for the robot, these would be perfect spheres, as the ends of a set of equal length arcs of circumference with a common origin engrench a circle. Since this is not exactly the case, the generated surfaces only resemble the sphericity predicted by the constant curvature model.

The addition of a second segment already generates a 4D workspace that is difficult to represent. The generation of this is a consequence of the fact that, from each point on the surface of the workspace of a segment, another similar surface is generated. The union of all of these surfaces, which arise from the points on the surface of the first segment, results in the two-segment workspace. This is a volume in which, in addition, each point can be reached from two different orientations, thus leaving latent the four degrees of freedom that PAUL would have with only two modules.

A similar reasoning can be applied to the three-segment case. Specifically, it has been observed, on the basis of the measurements carried out, that this can be framed in a volume of $200 \times 200 \times 100$ mm ($4$ dm$^3$), with the Euler angles varying, respectively, from 37 to 111°, from $-42$ to 40°, and from $-130$ to 57°.

### 5.3. Performance of the Table-Based Models

The size of the table to achieve acceptable kinematic modelling was set experimentally, as no previous references were available and previous works in the literature were very variable in terms of the number of data required. Furthermore, the possibility of an error occurring in the pneumatic system or the buffer of the vision acquisition system collapsing, together with the always present possibility of a leak in the segments, made it advisable to take data in small sessions and subsequently union of all of them. Since the data collection process was automated, this did not pose much of a problem.

Although the possibility that the ambient temperature was a factor that influenced the kinematics of the robot was considered during the dataset collection process, it was finally proven that small variations in temperature did not affect the behaviour.

Table 3 shows the datasets that were taken, the total time necessary to obtain them and the average time per point. It should be taken into account that not all captured positions were finally used, since, if the camera did not correctly detect the positions of the three beacon spheres, it could not calculate the orientation of the trihedral and therefore returned an error code. Of the 1200 samples collected, 5% had to be discarded, leaving 1146 finally usable. The average time per point, considering all the datasets collected, was 6.76 s, with a standard deviation of 0.63 s. The low variability between capture processes proves the effectiveness of the automated method designed.

**Table 3.** Time required to collect each dataset.

| Dataset Number | 1 | 2 | 3 | 4 | 5 | 6 | 7 |
|---|---|---|---|---|---|---|---|
| Number of Samples | 100 | 100 | 100 | 500 | 6 | 200 | 200 |
| Total Time (s) | 695 | 697 | 712 | 3884 | 38 | 1210 | 1199 |
| Time per Sample (s) | 6.95 | 6.97 | 7.13 | 7.77 | 6.47 | 6.05 | 6.00 |

Once all the datasets were combined, the direct and inverse kinematic models presented here were validated. The validation of the direct model consisted of sending the robot a combination of inflation times and measuring the distance between the position reached, captured by the cameras, and that predicted by the table. Repeating the experiment for 40 points, the histogram of results presented in Figure 15 was observed. The average error is 4.27 mm, the median error 2.72 mm, and the standard deviation 1.99 mm.

The high standard deviation and the shape of the histogram, tilted towards low values and with a very long tail, seem to indicate the existence of points where the model presents notable failures along with others with very good results. A future line of interest could be the detailed analysis of the workspace to locate where those regions of lower precision of the model are located and try to look for failures, perhaps leading to a greater density of points in the dataset.

In the same way, the inverse kinematic model was tested. To do this, PAUL was given a reference position and orientation to achieve, the necessary times were calculated, using the procedures referred to in Equations (15) and (16) inflation was carried out. Subsequently, the position captured with the cameras was compared with the desired one.

As expected, the existence of redundancies, in which equal position values are achieved with very different combinations of inflation times, introduces large uncertainties in the model, which the triangulation presented is not able to capture.

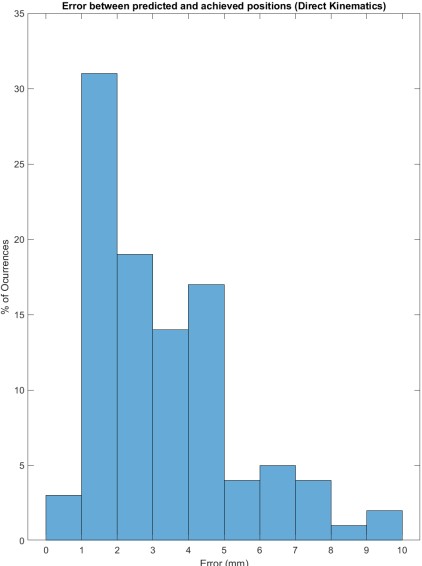

**Figure 15.** Histogram of the direct kinematic model experiment, which shows the distance between the point reached and the one predicted by the model. Source: authors.

Specifically, the inverse kinematic model has an average error of 10.78 mm, a median error of 9.22 mm and a standard deviation of 5.98 mm. While these errors may seem high, they are compared in Table 4 with other open-loop controllers presented in the literature. It can be clearly seen that they are in line with the results obtained and that they are even better than those obtained by smaller robots, where one would expect, due to the smaller working space, a higher accuracy (at least in data-driven models).

**Table 4.** Comparison of the results obtained by the inverse kinematics controller implemented on PAUL with other open-loop controllers in the literature. Only manipulators with more than one segment and open-loop controlled have been selected for the comparison. The average position error measured after moving the robot to a set of points is shown, where indicated as such in the paper.

| Reference | Actuation Type | Control Methodology | Robot Length | Error |
|---|---|---|---|---|
| [12] | SMA | FEM + FFNN | 240 mm | 4 mm |
| [67] | Tendon-driven | FEM | 1200 mm | 20 mm |
| [60] | Tendon-driven | Reinforcement Learning | 418 mm | 21.9 mm |
| [45] (3 segment, open-loop) | Pneumatic (HPN) | FFNN | 630 mm | 11.45 mm |
| [59] | Pneumatic (HPN) | Reinforcement Learning | 630 mm | 20 mm |
| [68] | Pneumatic (3D printed) | Reinforcement Learning | 400 mm | 22 mm |
| [56] | Pneumatic (STIFF-FLOP based) | FEM | 300 mm | 5.2 mm |
| PAUL | Pneumatic (STIFF-FLOP based) | Table-Based | 390 mm | 10.78 mm |

It is worth highlighting, however, two experiments in which PAUL performed very satisfactorily, because the area of operation was restricted to a region where no redundancies were found to exist. They are available in the video of Appendix A.

In the first of them, the robot was forced to reach a set of points located on the horizontal basis plane, also forcing the lower end of the last segment to be parallel to said plane. In all of them, errors less than 7 mm were achieved. Figure 16 shows the results of said experiment. With the aim of facilitating the understanding of the experiment, the beacon was changed for a laser pointer that points to the desired points, on which targets with a radius of 5 mm have been marked, allowing the accuracy achieved to be checked.

In the second experiment, shown in Figure 17, points on the lower horizontal plane were also taken as reference, but without imposing that the lower face of the robot should remain parallel to it. In this case, accuracies of 2 cm were achieved.

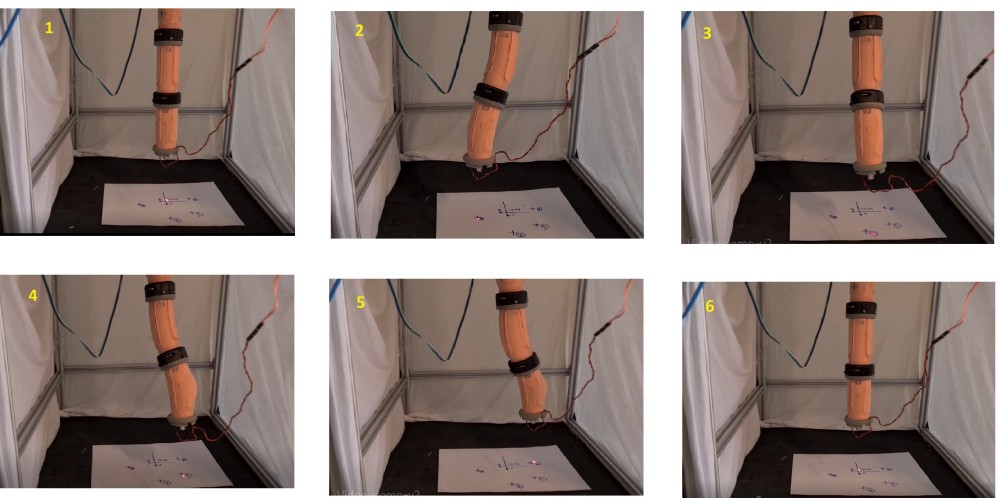

**Figure 16.** Results of Experiment 1, in which PAUL has been moved to different points located in the horizontal basis plane and forcing it to keep its lower end parallel to that plane. Beacon has been changed for a laser pointer to make experiment more understandable. Source: authors.

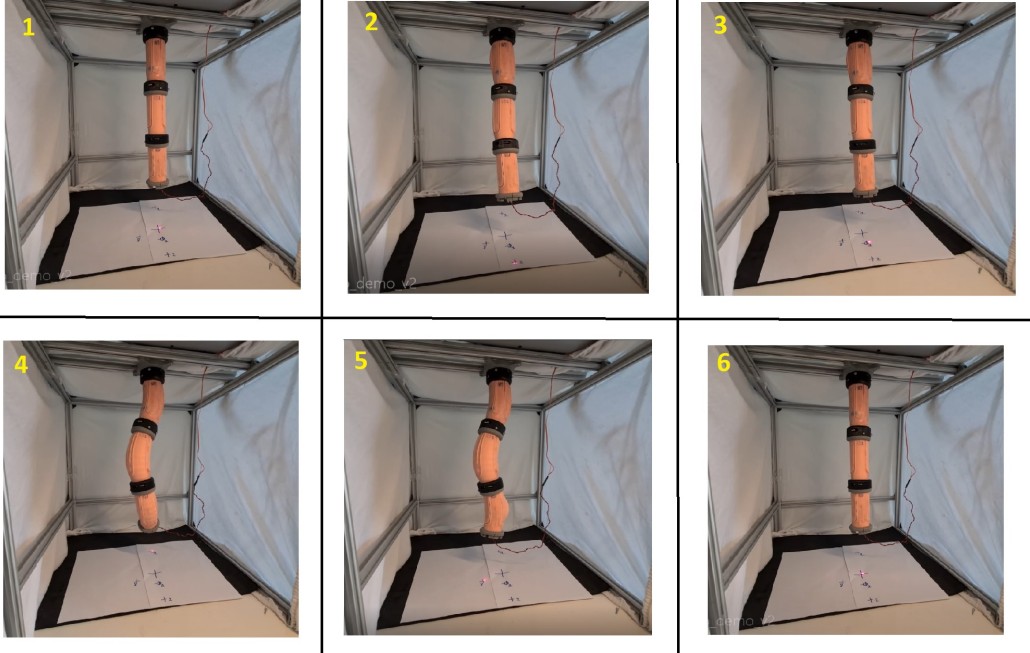

**Figure 17.** Results of Experiment 2, in which PAUL has been moved to different points located in the horizontal basis plane but without forcing it to keep its lower end parallel to that plane. Source: authors.

Although the inverse kinematic model therefore presents acceptable results, it must be commented that, in all these experiments, due to the geometry and material of the robot, at the moment in which PAUL reaches the desired position, it tends to acquire a movement damped oscillation. An attempt has been made to reduce it, despite everything, it is a very intrinsic phenomenon to the robot that is difficult to solve. A future line proposed, in this sense, is to try to rigidify the robot by introducing negative pressures that generate vacuum.

*5.4. Bending Experiments*

The first experiment consisted of analysing the deflection of a segment versus swelling time. For this purpose, one of the bladders was inflated continuously, in intervals of 100 ms. For each time, PAUL end coordinates $(x, y, z)$ are captured and the bending angle is calculated using the expression

$$\varphi = \arctan\left(\frac{\sqrt{(x - x_0)^2 + (y - y_0)^2}}{z}\right) \tag{17}$$

where $x_0$ and $y_0$ denote initial position of PAUL end.

Since the weight of the subsequent modules influences the behaviour of the first segment, the experiment was repeated by placing first one and then two additional segments. The results are shown in Figure 18.

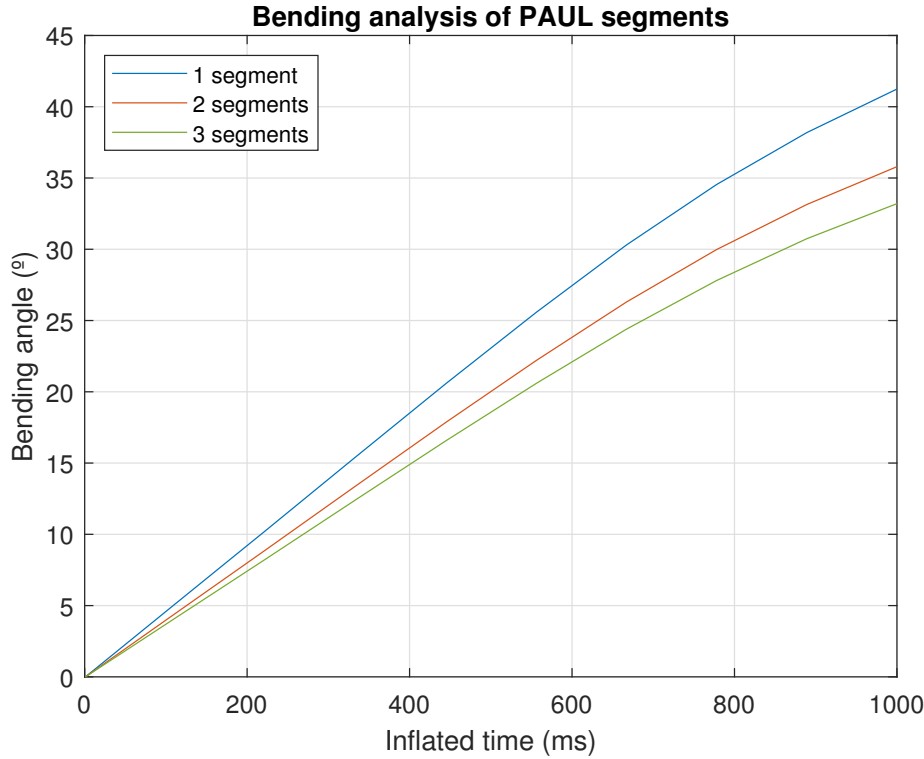

**Figure 18.** Bending achieved by one PAUL segment with different inflation times. Source: authors.

As can be seen, PAUL is capable of bending up to 40° to its vertical axis and the addition of new segments does not cause any noticeable decrease in its bending capacity. Although it remains far from the 80° in [28] or the 70° in [32], Pneunet segments and therefore more flexible, this is an acceptable bending capacity. Moreover, the fact that it does not substantially lose its bending capacity by adding segments makes it possible to concatenate bending movements and thus overcome obstacles that a rigid robot would not be able to overcome.

In conjunction with this, a validation test was proposed whose purpose was to demonstrate PAUL's ability to flex thanks to its deformable geometry. The aim was to point points in lateral planes. The results of this experiment are shown in Figure 19. The images, extracted from the video of Appendix A, show how the manipulator can adopt different shapes, is able to bend up to 40° and adapt, in case of obstacles, to a wide variety of geometries, which undoubtedly makes PAUL a fundamental ally in inspection and exploration operations in very cluttered environments.

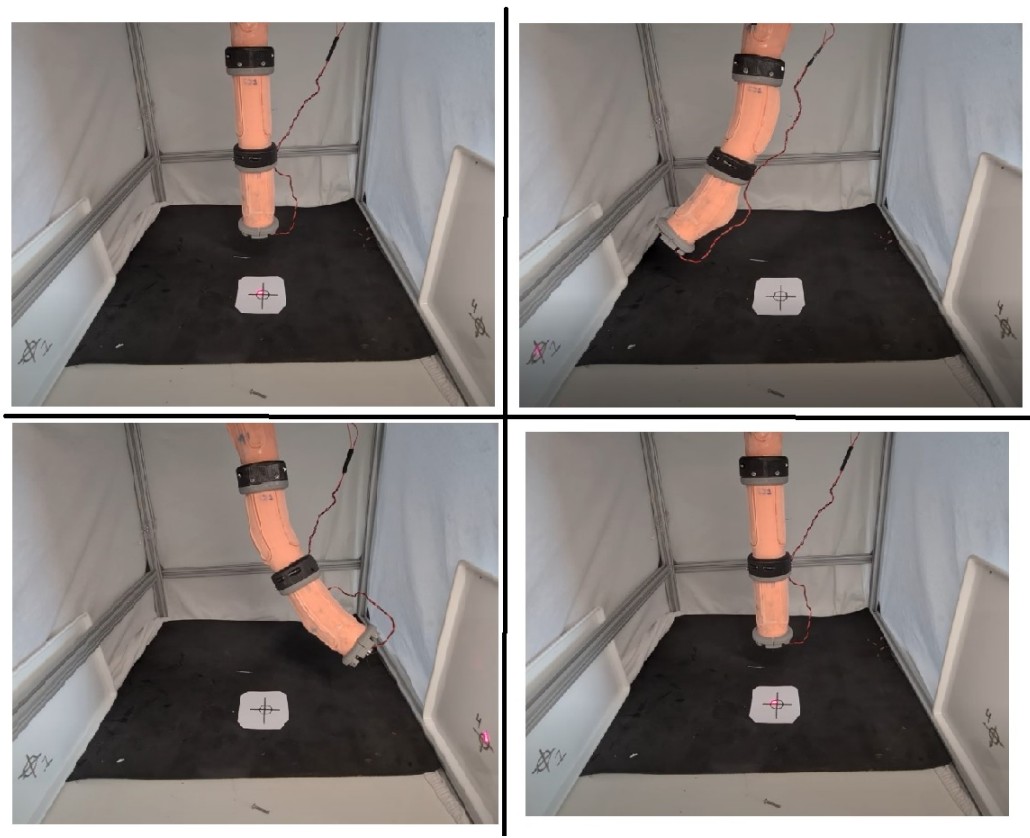

**Figure 19.** Second bending experiment, in which PAUL was asked to project his laser onto points on the side walls of the enclosing cube. Source: authors.

*5.5. Weight Carrying Experiments*

Finally, the load capacity of the robot and the performance of the kinematic model were evaluated and data was collected in vacuum, with different loads. For this purpose, an element similar to the joints between segments, also printed in PLA, was attached to the robot and different metal weights were placed on it. The device in question can be seen in Figure 20.

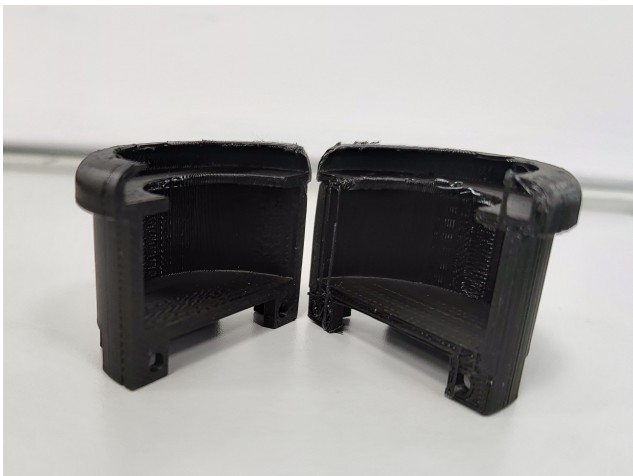

**Figure 20.** Element designed to keep the loads during the experiments carried out with weight. Source: authors.

The experiments consisted of taking PAUL to 10 different points in his workspace and comparing the position he reached with the position he would have reached if he had had no weight. The comparison was therefore made using the forward kinematic model, due to

its greater accuracy. Four different weights were tested: 55, 90, 130, and 155 g. These values are similar to those used in other works in the field for totally soft robot—excluding the hybrid ones, which, of course, have a much larger weight carrying capacity [47,61].

Figure 21 shows the results obtained. The mean errors are respectively 5.11, 4.40, 8.61 and 10.01 mm. From these data it can be deduced that, far from increasing progressively with weight, there are two categories: one that groups the experiments with loads of 55 and 90 g and another that groups those of 130 and 155 g. For the lower load values, the PAUL model predicts values with errors similar to those obtained by consulting the direct kinematics in the table obtained without weights. In the other cases, the results are clearly worse.

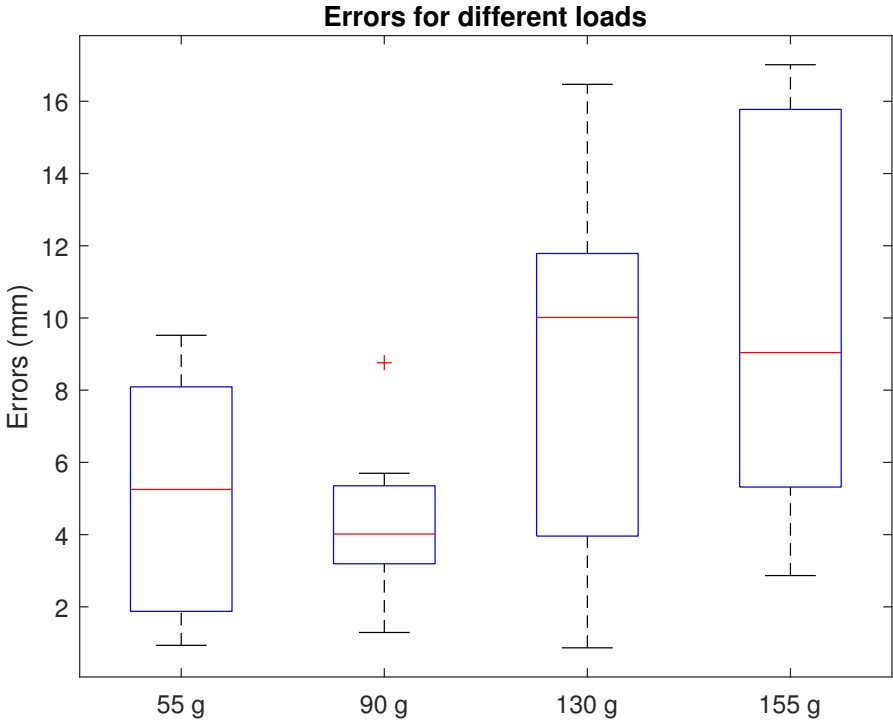

**Figure 21.** Errors between the point reached without weight and the point reached with a weight for different load values. The red line indicates the median error value, while the blue box includes all data between the first and third quartile values. Outliers are marked as addition signs. Source: authors.

It can also be seen that the lower error values, however, are quite similar in all cases. This is due to the fact that, at the points closest to the centre of its working space, the manipulator is able to position itself with much less error, regardless of the weight it is carrying. What the load produces, therefore, is not a systematic increase in error but a decrease in the working space, as PAUL lacks the strength to reach the areas furthest from the centre with the extra weight.

Table 5 compares PAUL performance those achieved in other works. Although there is room for improvement, the results obtained here are better than those of the closed-loop control of [47], where the average error obtained is 2 cm, and to those of the wired manipulator of [61]. While the latter seems to be independent of load, PAUL sees its error increase with increasing weight, which seems to indicate that, at higher weights, PAUL will indeed underperform.

Two aspects remain to be analysed. On the one hand, to see if between 90 and 130 g the increase in error is progressive or if, on the contrary, there is a point that clearly divides the two groups. On the other hand, it would also be necessary to study how an improvement

in the accuracy when empty would affect the accuracy when loaded: whether the robot would remain as accurate or whether these errors would not decrease.

**Table 5.** Loaded weights and errors achieved for different works in the literature.

| Work | Carried Weight | Error |
|------|---------------|-------|
| [47] | 105 g | 22 mm |
| [61] | 50 g | 15.8 mm |
| [61] | 15 g | 15.2 mm |
| PAUL | 50 g | 5.11 mm |
| PAUL | 90 g | 4.40 mm |
| PAUL | 130 g | 8.61 mm |
| PAUL | 155 g | 10.01 mm |

## 6. Conclusions

The numerous advantages and wide range of potential applications that soft robotics presents have made it a priority field of research in recent years. The difficulty, however, still present when it comes to both building and manufacturing robots, makes it a very incipient discipline. In the field of pneumatic robots, examples of existing manipulator arms are scarce, although cable and SMAs-based manipulators do exist.

In this work, PAUL has been presented, a modular robotic arm, made of silicone and that uses PLA for the joints between segments. Each module is made up of three bladders, which provide it with three degrees of freedom, which are reduced to two in the control in order to reduce redundancies. The actuation of the segments is done by sending different inflation times to each valve. The Pneunet type structure in the bladders allows the entire module to curve, reaching different positions in space.

The final implementation presented here has three segments and has had to address, first of all, an adequate design, which has been adjusted iteratively, and a correct selection of materials. Specifically, we have chosen a silicone that is slow enough to minimise bubbles, the main source of breakage and subsequent leaks, and at the same time, with sufficient hardness and density to avoid a high weight of the assembly. In addition to this, the pneumatic bench and the electronics in charge of controlling it had to be sized and designed.

PAUL has been modelled on open chain. To this end, a vision system has been designed, first of all, in charge of extracting, from the capture of a beacon, the position and orientation of the final end of the robot. Subsequently, an automated data collection process has been established that has allowed the generation of a dataset large enough to model, based on triangulations, both the direct and inverse kinematics of the system.

Specifically, accuracies of 4 mm have been obtained for direct kinematics and 11 mm in the best points for inverse kinematics, in line with existing results in the literature for manipulators with similar lengths (40 cm). In addition to this, the experiments have also demonstrated the enormous flexibility and bending capacity of PAUL as well as its ability to carry loads without increasing its positioning error, which reaffirms the ability of soft robotics to adapt to numerous applications.

Concretely, two applications have been considered where PAUL could have a very good fit. On the one hand, the inspection of pipes or unstructured environments with difficult access and twisted geometries. This is an application in which the position errors found here are negligible compared to the diameter of a pipe and in which, if necessary, the robot could be controlled by direct inflation of the bladders—making use of direct kinematics—since the aim is not to place PAUL at a certain point but to progressively sweep a region.

Its high modularity and its great capacity for adding segments make it possible to adapt to any type of environment to explore. Furthermore, given that it can carry small weights, the addition of a light camera (there are webcams weighing less than 100 g) would not represent any additional error. Although the bending capacity of this manipulator

is not the highest, the concatenation of bends in the different successive ones should be enough to adapt the shape of the pipe.

On the other hand, PAUL could be a useful aid in collaborative manipulation of light objects with humans. In various daily activities—such as in a warehouse, a pharmacy or a fast food establishment—the last part of the product delivery process consists of picking it up from a table and sorting it into drawers or rails. These are very light objects that are placed in spaces several centimetres wide. For optimisation reasons, many times the region where the products are classified is very high, that is, it would require a robot with bending capabilities of some importance.

Although there have been completely robotic solutions for years, due to the danger associated with rigid manipulators, these prevent workers from entering these areas. The use of soft robots, however, would allow collaborative work in them and reduce total costs, opening the door to automation for many small companies.

Great developments are therefore still expected in the coming years. Closed chain control, the use of more sophisticated modelling techniques and the management of inverse kinematics could be the contributions that PAUL would continue to make in this field in the near future.

**Author Contributions:** Conceptualization, A.B. and J.F.G.-S.; methodology, A.R.; software, A.R.; validation, A.R. and J.F.G.-S.; formal analysis, A.R., J.F.G.-S. and A.B.; investigation, A.R.; resources, A.B.; data curation, A.R.; writing—original draft preparation, J.F.G.-S.; writing—review and editing, A.B. and J.F.G.-S.; visualization, J.F.G.-S. and A.R.; supervision, A.B.; project administration, A.B.; funding acquisition, A.B. All authors have read and agreed to the published version of the manuscript.

**Funding:** Research activities were carried out at the Centre for Automation and Robotics, CAR (UPM-CSIC), within the Robotics and Cybernetics research group (RobCib), supported by the "Ayudas para contratos predoctorales para la realización del doctorado con mención internacional en sus escuelas, facultad, centros e institutos de I+D+i", funded by Programa Propio I+D+i 2022 from Universidad Politécnica de Madrid", by the TASAR (Team of Advanced Search And Rescue Robots)", funded by "Proyectos de I+D+i del Ministerio de Ciencia, Innovacion y Universidades" (PID2019-105808RB-I00), by the "RoboCity2030-DIH-CM, Madrid Robotics Digital Innovation Hub", S2018/NMT-4331, funded by "Programas de Actividades I+D en la Comunidad Madrid" and co-funded by Structural Funds of the EU, and by the "Proyecto CollaborativE Search And Rescue robots (CESAR)" (PID2022-142129OB-I00) funded by MCIN/AEI/10.13039/501100011033 and "ERDF A way of making Europe".

**Data Availability Statement:** The data and the code presented in this study are openly available at https://github.com/Robcib-GIT/PAUL (accessed on 3 January 2024).

**Conflicts of Interest:** The authors declare no conflicts of interest. The funders had no role in the design of the study; in the collection, analyses, or interpretation of data; in the writing of the manuscript; or in the decision to publish the results.

## Abbreviations

The following abbreviations are used in this manuscript:

| | |
|---|---|
| DOF | Degrees of Freedom |
| EAP | Electroactive Polymer |
| FEM | Finite Elements Method |
| FFNN | Feedforward Neural Network |
| HPN | Honeycomb Pneumatic Network |
| ML | Machine Learning |
| MSER | Maximally Stable External Regions |
| PAM | Pneumatic Artificial Muscle |
| PAUL | Pneumatic Articulated Ultrasoft Limb |
| PCC | Piecewise Constant Curvature |
| SMA | Shape Memory Alloys |
| TCA | Twisted and Coiled Actuators |

## Appendix A. Conducted Experiments

A video presenting the validation of the inverse kinematic model and PAUL's bending ability can be consulted in the following link: https://www.youtube.com/watch?v=1XM6 AjTwlqs (accessed on 3 January 2024).

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
