# Peer review of "Design, Manufacturing, and Open-Loop Control of a Soft Pneumatic Arm"

_actuators, doi:10.3390/act13010036_

Round 1
Reviewer 1 Report
Comments and Suggestions for Authors
The paper presents the design, manufacturing and open-loop control of a 6-degrees of freedom (DOF) soft pneumatic arm called PAUL. The proposed solution for the soft arm is interesting, it has applicability potential, but the manner of presentation is wanting.
The authors may consider the following comments for revising the paper:
· The general drafting of the paper is unsatisfactory. The location of the figures in the body of the paper is not adequate, the figures are at a distance from the explanations that refer to them, and consequently the entire paper is difficult to follow.
· An thorough revision of the English language used in the paper is required.
· Line 114: 44 kg.
· Line 192: depicted.
· Figure 3 and Table 1 should be put together, in other words the dimensions should be indicated directly on the drawing.
· The titles of the tables should be positioned above the tabels.
· Please explain: How are the issues related to hysteresis solved?
· Considering the reduced positioning precision of the proposed construction, please name possible applications of this equipment.
Comments on the Quality of English LanguageExtensive editing of English language required
Author Response
The paper presents the design, manufacturing and open-loop control of a 6-degrees of freedom (DOF) soft pneumatic arm called PAUL. The proposed solution for the soft arm is interesting, it has applicability potential, but the manner of presentation is wanting.
First of all, we would like to thank you for your detailed reading of the article and your valuable comments which have helped us to improve the paper. Your suggestions have been implemented and now appear highlighted in the manuscript. Explanations of how it has been done can be find bellow.
The authors may consider the following comments for revising the paper:
The general drafting of the paper is unsatisfactory. The location of the figures in the body of the paper is not adequate, the figures are at a distance from the explanations that refer to them, and consequently the entire paper is difficult to follow.
A drafting reorganization has been carried out. Now all the Figures and Tables appear only one or two paragraphs after their first reference in the text. The main problem was Figure 5, which, because it was placed at the end of 3.3 section, “blocked” the rest of the Figures. Now, it has been placed at the beginning of the section, so the rest of the Figures can be displayed just after the place they are firstly referenced. The only exception is Figure 10, which cannot be placed higher due to the excess of images, all of them of considerable size, in that part of the manuscript.
An thorough revision of the English language used in the paper is required.
Line 114: 44 kg.
Line 192: depicted.
Both typos have been corrected. An extensive revision has been done looking for other typos. Additionally, a general revision of English have been done.
Figure 3 and Table 1 should be put together, in other words the dimensions should be indicated directly on the drawing.
Following your comments, Table 1 has been removed and all the dimensions are now indicated in Figure 3, which has been redesigned.
The titles of the tables should be positioned above the tables.
Thank you very much for the observation. They have been correctly placed.
Please explain: How are the issues related to hysteresis solved?
Final part of Section 4.3 has been reworded to make it clearer. Specifically, the paragraph concerning hysteresis now explains how it was addressed is included in this response:
Finally, there are hysteresis phenomena in the silicone that cause the position reached by inflating for a time t to be different from the position reached by inflating first for a time t1 and then for a time t2 = t – t1. The strategy employed to tackle this problem was to capture the dataset bringing PAUL back to its zero position between each sample. Nevertheless, when controlling the robot in open-loop this is not possible, or, at least, not desirable, as one may wish to follow trajectories or travel through a sequence of points. Therefore, transitioning from position x1 to x2 requires an additional factor of 1.2, also derived experimentally, to account for hysteresis effects.
Considering the reduced positioning precision of the proposed construction, please name possible applications of this equipment.
In the conclusion, two possible applications of PAUL have been proposed and justified:
- Inspection of pipes and unstructured spaces
- Sorting objects in warehouses or fast food stores
Final comment:
Many thanks again for reading the article and the various points made. We have tried to pay more attention to language, beyond the aforementioned possibility of hiring the editing service. The comment related to applications has helped us to think about what our work can contribute to society, beyond its purely scientific value. This is an aspect that has enriched our article, since engineering must never forget its social vocation
Reviewer 2 Report
Comments and Suggestions for Authors
The authors present the design, manufacturing and open-loop control of a polymeric pneumatic continuum robot composed of three 100 mm long segments. I think the following points should be clarified.
1) The design seems to already exist and here is a minor variation. The open-loop control has a pretty large error and does not consider payloads or external forces such that it cannot be said to be a strong contribution. Can the authors specify what their contribution to science is?
2) The authors state "gives room to work with loads of a certain magnitude" but no payload experiments are shown. I also see a criticism of previous work stating that "it is not possible to freely fix the position and orientation of the final tip of the manipulator" but would like to note that the present arm is not able to fix the orientation of the final tip because it lacks a twisting DOF.
3) The authors make some confusing statements about DOFs. They say that three bladders leads to three degrees of freedom. But the arm itself seems to bend in two directions so it has two degrees of freedom. There are multiple such statements throughout, but DOFs should refer to the possible types of motion of the arm, not the number of inputs to the segment. Otherwise a segment with 10 bladders would have 10 DOFs.
4) There are several critical issues with the literature review. I believe that the authors are not very familiar with the field and are making various mistakes:
a) PAM generally refers to pneumatic artificial muscles not pneumatic actuated muscles.
b) The authors attribute PneuNet to a 2019 paper. The term originates from, I believe, the famous 2014 paper "Pneumatic Networks for Soft Robotics that Actuate Rapidly".
c) The literature review on pneumatic actuators that begins with "According to [44]" says "of all the actuation possibilities" but then lists a very small subset of pneumatic actuators considering the extremely large range of pneumatic actuators that do not use PAMs or PneuNets for linear, bending, or multi-DOFs actuation. A more informed discussion of the literature is necessary.
d) The authors list too few pneumatic robotic arms. Some works such as those using McKibben actuators "Soft Assistive Robot for personal care of elderly people" followed by numerous control papers including "Model Based Reinforcement Learning for Closed Loop Dynamic Control of Soft Robotic Manipulators" are entirely soft and should be included. Hybrid hard-soft approaches such as "Design of a lightweight soft robotic arm using pneumatic artificial muscles and inflatable sleeves", "Toward the development of large-scale inflatable robotic arms using hot air welding" and "Torsional Pneumatic Actuator Based on Pre-Twisted Pneumatic Tubes for Soft Robotic Manipulators". Overall, I am seeing a paper that is missing most of the relevant literature yet contains many unneeded references.
Comments on the Quality of English LanguageOverall fine. Another reading could improve the quality of language.
Author Response
The authors present the design, manufacturing and open-loop control of a polymeric pneumatic continuum robot composed of three 100 mm long segments. I think the following points should be clarified.
Thank you very much for the detailed reading of the article, in particular for the depth of the comments regarding the bibliography, which have helped us to enrich our manuscript. We have implemented all your suggestions. In the following lines, we respond to the various comments that have been brought to our attention.
1) The design seems to already exist and here is a minor variation. The open-loop control has a pretty large error and does not consider payloads or external forces such that it cannot be said to be a strong contribution. Can the authors specify what their contribution to science is?
The paragraph where main contribution is presented has been enlarged:
The main contribution of this work has been the design, manufacturing and open-loop control of a 5-degrees of freedom (DOF) soft pneumatic arm called PAUL (Pneumatic Articulated Ultrasoft Limb) and depicted in Figure [1] capable of carrying light loads without increasing its precision error. In addition to the precision of its control system without and with external payloads, its workspace and its bending capacity are also analysed. It is not common to find a simultaneous analysis of all these factors in the literature.
Hence, we introduce a manipulator akin to existing models, thoroughly examined and confirmed to deliver commendable performance across various aspects of interest—such as working space, flexibility, and precision—without exceptional superiority in any one aspect. However, it notably excels in load capacity, rendering it well-suited for specific applications that do not require pushing any particular characteristic to its limits.
2) The authors state "gives room to work with loads of a certain magnitude" but no payload experiments are shown. I also see a criticism of previous work stating that "it is not possible to freely fix the position and orientation of the final tip of the manipulator" but would like to note that the present arm is not able to fix the orientation of the final tip because it lacks a twisting DOF.
A new section describing experiments with weights has been added. These are not large weights but they are comparable to those of other soft robots in the literature (comparison is established inside this section). Hybrid manipulators have been omitted from this comparison as they have a much higher load capacity due to their stiffness. However, possible applications linked to the results of these experiments are now discussed in the Conclusions.
With regard to the degrees of freedom, the robot does indeed have 5 and not six degrees of freedom, and this has been corrected in this new version, as shown in line 36.
In addition, the comment on Alessi's work has been removed. This, rather than a criticism, was intended to draw some distinction between "arm" and "segment".
3) The authors make some confusing statements about DOFs. They say that three bladders leads to three degrees of freedom. But the arm itself seems to bend in two directions so it has two degrees of freedom. There are multiple such statements throughout, but DOFs should refer to the possible types of motion of the arm, not the number of inputs to the segment. Otherwise a segment with 10 bladders would have 10 DOFs.
Everything related to degrees of freedom has been reformulated in order to make it more rigorous and understandable and to correct errors and inaccuracies.
In particular, the paragraph describing the degrees of freedom of a segment has been rewritten to explain that there are three degrees of freedom per segment (inflating its three bladders implies lengthening it):
PAUL is composed of 3 independent modules or segments, made of silicone. Each segment is fed by 3 pneumatic tubes whose inflation give him 3 degrees of freedom: it can bend in two direction, as well as elongate when the three bladders are inflated. To reduce redundancies, it was decided to act only up to two at a time.
In addition, all sentences where the term degrees of freedom was incorrectly used to refer to robot inputs have been corrected. In addition to the word inputs itself, the terms bladders or valves are used for inputs in the configuration space of the robot. It can therefore be said that PAUL, as a whole, has 9 independent inputs, which give it, due to its structure and design, 5 degrees of freedom. No more than 6 inputs are ever actuated simultaneously.
4) There are several critical issues with the literature review. I believe that the authors are not very familiar with the field and are making various mistakes:
a) PAM generally refers to pneumatic artificial muscles not pneumatic actuated muscles.
Thank you very much for the warning, we have corrected the error.
b) The authors attribute PneuNet to a 2019 paper. The term originates from, I believe, the famous 2014 paper "Pneumatic Networks for Soft Robotics that Actuate Rapidly".
Citation has been corrected. No other previous work using the term Pneunet (or pneu-net as it was written in the paper) has been found.
c) The literature review on pneumatic actuators that begins with "According to [44]" says "of all the actuation possibilities" but then lists a very small subset of pneumatic actuators considering the extremely large range of pneumatic actuators that do not use PAMs or PneuNets for linear, bending, or multi-DOFs actuation. A more informed discussion of the literature is necessary.
(Answered with the following comment)
d) The authors list too few pneumatic robotic arms. Some works such as those using McKibben actuators "Soft Assistive Robot for personal care of elderly people" followed by numerous control papers including "Model Based Reinforcement Learning for Closed Loop Dynamic Control of Soft Robotic Manipulators" are entirely soft and should be included. Hybrid hard-soft approaches such as "Design of a lightweight soft robotic arm using pneumatic artificial muscles and inflatable sleeves", "Toward the development of large-scale inflatable robotic arms using hot air welding" and "Torsional Pneumatic Actuator Based on Pre-Twisted Pneumatic Tubes for Soft Robotic Manipulators". Overall, I am seeing a paper that is missing most of the relevant literature yet contains many unneeded references.
Based on your comments, all the references in the article have been revised, the existing literature has been thoroughly reviewed and the Related Works section has been completely restructured. The first step was to describe the different types of pneumatic actuation, which have been classified as follows:
- Bio-inspired actuation
- PAM and derivatives
- Pneunet and derivatives
- Jamming-based actuators
Subsequently, examples of pneumatic actuation arms were presented. They have been presented:
- Hybrid soft-rigid arms
- Origami arms
- 3D printed arms
- HPN actuation based arms
- Arms based on the STIFF-FLOP manipulator
All articles you comment on are now cited. Manti et al article “Soft Assistive Robot for personal care of elderly people”, although already cited in the introduction under soft robotics applications, is now also discussed in this state-of-the-art section.
Final comment:
Thank you again for your time and comments on the article. They have helped us to learn more about the literature, to structure the article better and to reflect on what PAUL brings to the current soft robot landscape. The various suggestions for further research have led to the addition of two experiments that we consider valuable and which we hope you will find of interest.
Round 2
Reviewer 1 Report
Comments and Suggestions for Authors
Almost all of my comments were taken into consideration.
Comments on the Quality of English LanguageMinor editing of English language required
Reviewer 2 Report
Comments and Suggestions for Authors
The authors have addressed all of my comments and I recommend the paper for publication in its current form.